# Risk assessment of sea ice disasters on fixed jacket platforms in the Liaodong Bay

Ning Xu[1], Shuai Yuan[1], Xueqin Liu[1], Yuxian Ma[1], Wenqi Shi[1], Dayong Zhang[2]

[1]National Marine Environmental Monitoring Centre, Dalian, 116023, China

[2]School of Ocean Science and Technology, Dalian University of Technology, Panjin, 124221, China

*Correspondence to*: Ning Xu (11229585@qq.com)

**Abstract.** Sea ice disasters seriously threaten the structural safety of oil platforms in the Bohai Sea. Therefore, it is necessary to carry out the risk assessment of sea ice disasters on oil platforms in the Bohai Sea. In the study, a risk assessment of sea ice disasters on fixed jacket platforms in the Liaodong Bay, Bohai Sea was performed in five steps. Firstly, the formation

mechanisms of sea ice disasters were analyzed and the sources and modes of sea ice risks were summarized. Secondly, according to the calculation formulas of extreme ice force, dynamic ice force and accumulated force, several ice indices such as thickness, motion, strength, period, and concentration were proposed as the hazard indices and corresponding values were then assigned to the proposed indices based on ice conditions in the Bohai Sea. Thirdly, based on four structural failure modes (structural overturning by extreme ice force (Mode 1), structural fracture failure caused by dynamic ice force (Mode 2), the

damage of facilities caused by dynamic ice force (Mode 3), and structural function failure caused by accumulated ice (Mode 4)), the structural vulnerability index, overturning index, dynamic index, ice-induced vibration index, and function index were proposed and corresponding values were assigned to the structural vulnerability index of fixed jacket platforms in the Liaodong Bay. Fourthly, the weight of each risk index was determined according to the previously recorded sea ice disasters and accidents and the sea ice risk was then calculated with the weighted synthetic index method. Finally, with the above index

system and risk assessment methods, the risk assessment of sea ice disasters on 10 jacket platforms in three sea areas in the Liaodong Bay was carried out. The analysis results showed that efficient sea ice prevention strategies could largely mitigate the sea ice-induced vibration-related risks of jacket platforms in the Liaodong Bay. If steady-state vibration occurs (usually in front of the vertical legged structure) or the structural fundamental frequency is high, the structural vulnerability is significantly increased and the calculated risk levels are high. The sea ice risk assessment method can be applied in the design, operation,

and management of other engineering structures in sea ice areas.

## 1 Introduction

The optimization selection and safety assessment of engineering structures generally involve the long-term simulation calculation and require a certain calculation basis. If an assessment index system is established based on the risk assessment

theory, the safety level of various structures and the risk level of structural operation can be quickly assessed. Therefore, a structural risk assessment index system can be used as a basis for the more efficient and accurate structural safety analysis.

In recent years, the losses caused by sea ice disasters have increased significantly (Fang et al., 2017). In China, sea ice disasters mainly occur in the Bohai Sea and the North Yellow Sea. Sea ice can push down offshore platforms, destroy ships and offshore engineering facilities, impede navigation, and cause losses to offshore and tidal aquaculture (Zhang et al., 2013; Wang et al., 2011; Lu, 1993; Ding, 2000). In the ice period of 1969, the entire Bohai Sea was covered by sea ice and ice thickness even reached 60 cm. In the sea ice disaster, the No. 2 living and drilling platforms collapsed due to the huge thrust of sea ice, thus leading to a great impact on the economy of China. In 1977, the beacon tower of No. 4 drilling well was also pushed down by sea ice. On January 28, 2000, the JZ20-2 MS Platform suffered the severe steady-state vibration under the action of level ice, thus causing the fatigue fracture of the evacuation pipeline of the No. 8 well, natural gas leakage, and platform suspension (Yue et al., 2009; Li et al., 2008; Timco et al., 2011). Since 2010, the aquaculture area affected by sea ice disasters has reached 40,000 hectares per year (State Oceanic Administration, 2011-2016).

Since the 1980s, Chinese scholars studied the preventive measures of sea ice disasters (Ouyang et al., 2017; Lu et al., 1993; Wang et al., 2011; Zhang et al. 2015), sea ice measurement and forecast (Luo et al., 2004; Zhao et al., 2014; Su and Wang, 2012), engineering coping strategies (Zhang et al., 2010, 2016), and mechanisms (Yue et al., 2009; Li et al., 2008, Liu et al., 2009; Huang and Li, 2001; Wang et al., 2018; Yue et al., 2007a). Most studies on existing sea ice risk assessment only involved the descriptions of sea ice. Guo et al. (2008) proposed three sea ice parameters including thickness, strength and period as the influencing factors of sea ice disasters and established three sea ice disaster risk levels, such as zero risk, low risk, and high risk. Gu et al. (2013) converted sea ice thickness into a sea ice hazard index and determined sea ice hazard risk levels. However, due to the differences in the classification results of sea ice disasters on different offshore engineering structures and main sea ice factors, the sea ice data required in the assessment of structural ice disaster are different. Therefore, previous results cannot meet the current engineering requirements.

Risk assessment studies of engineering structures under environmental loads are mainly focused on large-span bridges, houses and other buildings under wind and seismic loads (Park et al., 1985; Schiff et al., 1994; Kameshwar et al., 2014), and corresponding vulnerability curves were obtained (Hwang and Liu, 2004; Singhal and Kiremidjian, 1996; Khanduri and Morrow, 2003). Risk assessment of sea ice loads on marine platforms, risk assessment index or method was seldom reported. The fundamental studies are mainly focused on ice force calculation methods (Sanderson, 1988; Ou et al., 2002), structural failure mode analysis (Yue et al., 2008), ice force resistance of engineering structures (Wang et al., 2012), and fatigue life calculation (Li et al., 2008; Liu et al 2006). The previous results might be used as the basic theory for establishing a sea ice risk assessment index system and assessment method of marine structures.

This paper focuses on the risk assessment methods of sea ice disasters on jacket platforms. The hazard indices of sea ice were firstly determined based on the forms of sea ice force on platform structures. Then, the weight coefficients of these indices were calculated with ice force calculation formulas. Then, the physical vulnerability index was determined according to the

platform failure modes and the weight of the vulnerability index was determined based on the previously recorded sea ice disasters and accidents. Sea ice disaster risks on 10 jacket platforms in three sea areas were individually assessed with the overall risk assessment method and the multi-mode risk assessment method. The assessment results indicated that except that several auxiliary platforms are in the high risk level, other platforms are in the healthy condition, but safety management in winter should be further enhanced.

## 2 Study area

The Bohai Sea is a seasonal ice-covered sea in the latitude range from 37° N to 41° N. It is also the ice-covered sea with the lowest latitude in the northern hemisphere. Liaodong Bay in the Bohai Sea is the most severely icy bay with an ice period of about 130 days. The edge of the ice-covered region is near the contour line of 15 m and about 70 nautical miles away from the top of the bay. Generally, ice thickness in the Bohai Sea is 30~40 cm. The sea ice drifting speed is generally 0.5 m/s and the maximum speed is about 1.5 m/s. The dominant wind in winter is northerly wind. Due to the clockwise flow along the coast and the right-turning tide, sea ice conditions in the east are more serious than those in the west. In addition, the warm current from Yellow Sea flows through the northern Bohai Strait into the Bohai Sea, and then to the west bank of Liaodong Bay roughly along the northwest direction, thus raising the water temperature in the western part of the Liaodong Bay. Therefore, sea ice conditions in the west of the Liaodong Bay are heavier than those in the east.

Based on the distribution characteristics of sea ice in China, the Bohai Sea and the North Yellow Sea were divided into 21 regions in such a way that the ice conditions in each region were basically the same. Then, the design parameters of marine structures in the 21 regions areas were proposed, including physical and mechanical parameters as well as key parameters of ice conditions (period, thickness and motion). The sea ice parameters in each ice region provide a useful reference for the design of ice structures and the fatigue life assessment of existing structures.

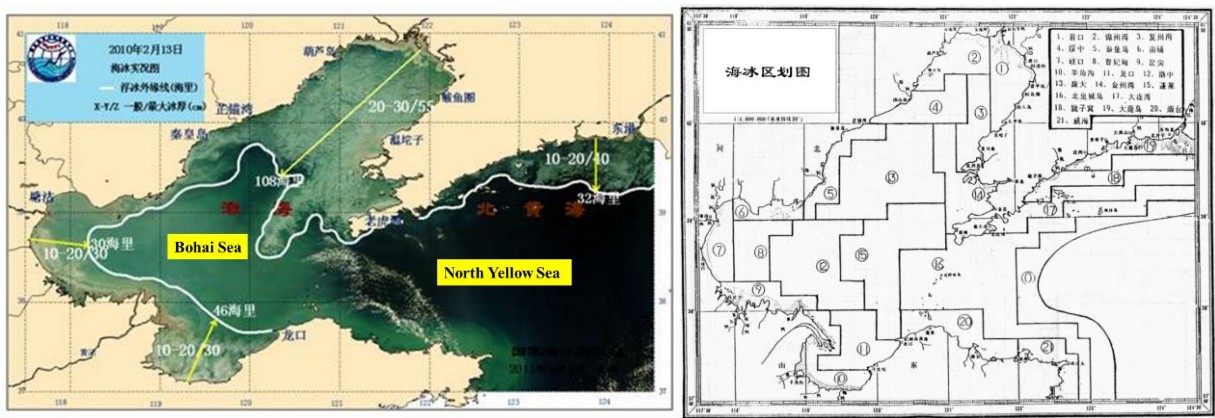

**Figure 1: Location map of Bohai Sea and North Yellow Sea. (State Oceanic Administration, 2010; China National Offshore Oil Corporation, 2002)**

Oil platforms in Bohai Sea have two structural categories: caisson structure and jacket structure. The latter is the dominant structure. In addition to the multi-legged structures (usually 4 legs, as shown in Fig. 2(a)), the single-legged structure has been widely applied in auxiliary platforms (Fig. 2(b)) and even main platforms (Fig. 2(c)). In oil platforms, ice-breaking cones are generally adopted to reduce the impact of sea ice. Old platforms had been equipped with ice-breaking cones (Fig. 2(d)) and new platforms were designed as the cone category (Fig. 2(e)). Due to the differences in ice conditions, structural category, dynamic performance, function and structural ice resistance, platform structures in the Liaodong Bay showed significant differences in sea ice risk levels. Current sea ice management measures in winter effectively reduced sea ice risks in oil and gas exploitation.

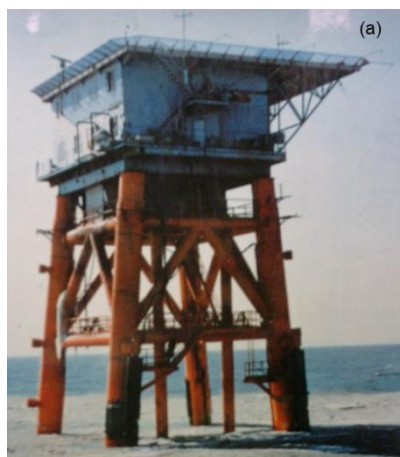 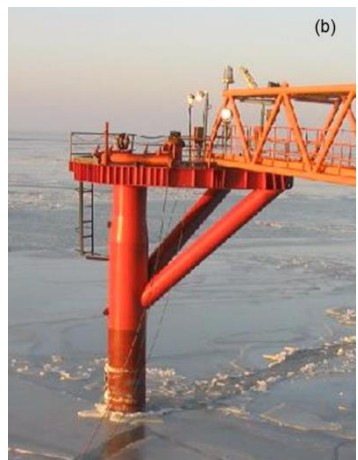 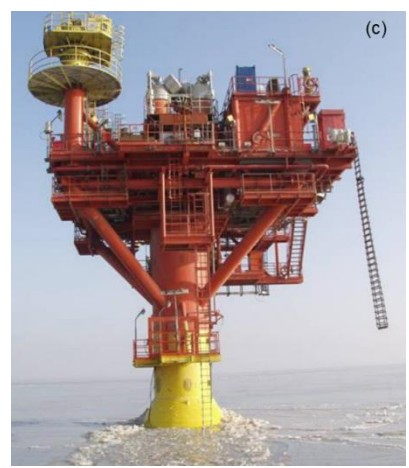

**(a) Four-legged oil production platform (built in 1987); (b) One-legged auxiliary platform (built in 1999); (c) One-legged oil production platform (built in 2003).**

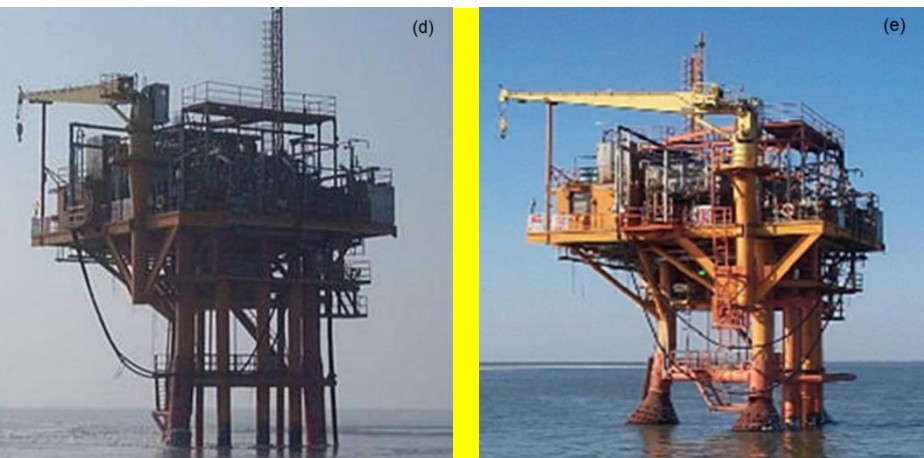

**(d) Three-legged upright pile production platform (built in 1997); (e) Three-legged upright pile production platform with cones (built in 2000)**

**Figure 2: Main structural forms of jacket platforms in the Bohai Liaodong Bay.**

Oil platforms are densely distributed in the Liaodong Bay, especially the narrow form of jacket structures. The impact of sea ice is significant in the Liaodong Bay. Therefore, it is necessary to carry out the risk assessment of sea ice disasters on jacket oil platforms in the Liaodong Bay.

## 3 Research methods

### 3.1 Technical routes

With the risk assessment method of natural disasters (Zhang and Li, 2007; Tachiiri, 2012), the technical routine of the risk assessment of sea ice disasters on oil platforms was established below (Fig. 3).

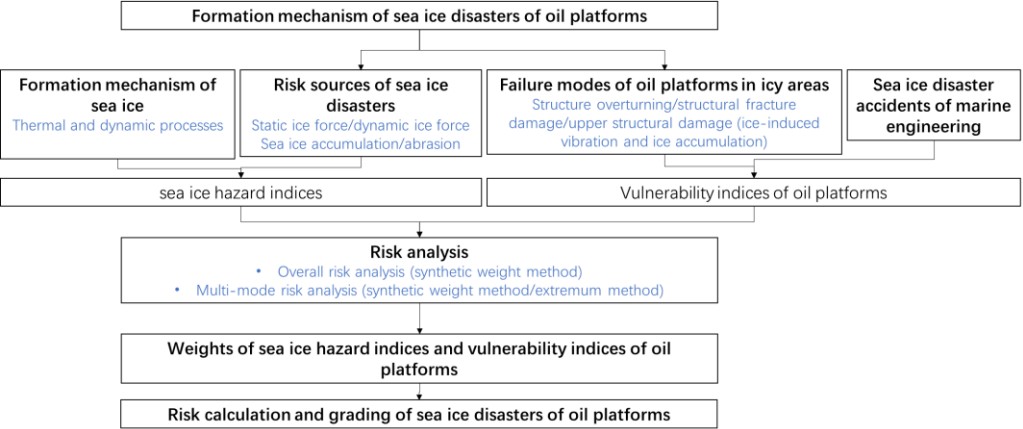

**Figure 3: Flow chart of risk assessment of sea ice disasters on oil platforms.**

### 3.2 Risk assessment model index system

With the synthetic index method (Zhang and Li, 2007), the sea ice risk assessment index system of oil platforms was established with the risk model (Risk = $F$ (Hazard, Vulnerability, Resistance)) to calculate the risk level. Firstly, the hazard index ($H$), vulnerability index ($V$), and resistance index ($R$) were graded and corresponding values were assigned to these indices. Then the sea ice hazard index was qualitatively described as 5 levels: extremely high, high, medium, low, and extremely low and the corresponding quantitative values were respectively set to be 5, 4, 3, 2, and 1. The structural vulnerability index and the resistance levels were described qualitatively as 3 levels: high, medium and low and the corresponding values were set to be 5, 3, and 1, respectively.

Sea ice disasters on oil platforms have different risk modes. Therefore, in the establishment of risk assessment index systems and evaluation models, two methods can be adopted: overall risk assessment and multi-mode risk assessment. With the overall risk assessment method, the weights of secondary indices are determined to calculate the hazard index ($H$), vulnerability index ($V$), and resistance ($R$) and then the overall risk $I_e$ is calculated according to Eq. (1). With the multi-mode risk assessment method, the assessment results of various risk modes $I_{s,i}$ are determined with the hazard index $H_i$, the vulnerability index $V_i$,

and the disaster resistance ability index $R_i$ of various risk modes (Eq. (2)). Then, the risk level with the highest risk level $I_{s,max}$ can be calculated according to Eq. (3)). Finally, with the cumulative weight coefficient of various risk modes, the overall risk $I_e$ or the highest risk level of various modes $I_{s,max}$ can be selected.

$$I_e = HVR = \sum \omega_i H_i \sum \omega_j V_j \sum \omega_k R_k \ , \tag{1}$$

$$I_s = \sum \lambda_i I_{s,i} \ , \tag{2}$$

$$I_{s,max} = max I_{s,i} \ , \tag{3}$$

where $I_e$ is the overall risk assessment result; $H$, $V$, and $R$ are respectively the overall hazard, vulnerability, and disaster resistance indices; $i$, $j$, and $k$ are respectively the numbers of $H$, $V$, and $R$ secondary indices; $H_i$, $V_j$, and $R_k$ are the secondary indices; $\omega_i$, $\omega_j$, and $\omega_k$ are respectively the weight coefficients of these secondary indices; $I_s$ is the calculation result of the multi-mode risk assessment method; $I_{s,max}$ is the calculation result with the maximum risk value according to the multi-mode risk assessment; $I_{s,i} = H_i V_i R_i$ is the calculation result of the $i$-th risk mode; $\lambda_i$ is the weight of the $i$-th risk mode.

## 4 Establishment of the risk assessment index system of sea ice disasters

### 4.1 Formation mechanisms of sea ice disasters

The impact of sea ice is the main cause for accidents and risks of marine structures in ice-covered areas. The impact energy mainly comes from wind, currents, thermal expansion and sea ice (Sanderson, 1988). Sea ice disasters have three major risk sources (extreme ice force, dynamic ice force, and sea ice accumulation) and four structural risk modes (structural overturning by extreme ice force, structural fatigue damage induced by ice vibration, upper facility damage caused by ice vibration, and facility damage caused by ice jamming) (Zhang et al, 2015).

#### 4.1.1 Risk sources of sea ice disasters and major sea ice risk factors

4.1.1.1 Extreme ice force

When sea ice contacts a structure, sea ice exerts a relatively stable ice force on the structure. Extreme ice force generates static effects and transient impacts. In the sea areas around oil platforms in the Liaodong Bay, the sea ice flow rate can reach 1.4 m/s and extreme ice force is directly affected by sea ice thickness, sea ice strength and structural width. The calculation methods of static ice force are given in various engineering design standards of cold zones. For example, when ice is crushed in front of a structure, the ice load generated on the structure can be calculated as follows (API, 1995):

$$F_c = mI f_c \sigma_c Dt, \tag{4}$$

where $F_c$ is extrusion ice load; $m$ is shape factor and respectively set as 0.9, 1.0 and 0.7 for circular section, square section with a positively applied ice load and square section with an obliquely applied ice load; $I$ is the embedding coefficient; $f_c$ is

the contact coefficient; $\sigma_c$ is the uniaxial compression strength of level ice in the horizontal direction, MPa; $D$ is structural width; $t$ is the designed level ice thickness.

The horizontal component of ice bending load applied on a slope structure is:

$$F_H = K_\alpha t^2 \sigma_f tan\beta, \tag{5}$$

where $F_H$ is the horizontal component of ice bending load applied on a slope structure; $K_\alpha$ is the coefficient related to the structure and ice thickness; $t$ is the designed level ice thickness; $\sigma_f$ is bending strength of level ice; $\beta$ is the angle (°) between the structural slope and the horizontal plane (RIL, 2001).

4.1.1.2 Dynamic ice force

When sea ice continuously passes through a structure, it generates a periodic impact load on the structure, namely, dynamic ice force. Dynamic ice force usually occurs on narrow structures. Under the action of dynamic ice force, a structure undergoes vibration, namely, ice-induced vibration. When the frequency of dynamic ice force is consistent with the structural frequency, it causes strong vibration due to resonance. The ice force period is calculated as (Qu et al., 2006):

$$T = l_b/v, \tag{6}$$

where $T$ is ice force period; $l_b$ is breaking length of ice plate and affected by ice thickness, ice strength, structural diameter, and ice velocity; $v$ is ice velocity.

4.1.1.3 Sea ice accumulation

After floe is applied on a structure and then broken, if broken ice is not removed in time due to structural blockage, ice accumulates. Usually, broken ice accumulates in front of a wide structure or in the vicinity of dense ice-collecting members, such as the pile leg of an oil platform and the coarse grid at the water intake port of a nuclear power plant. The growth, size, and load of accumulation ice had been extensively explored. The calculation methods of extreme ice force in various specifications also involve ice accumulation. Sea ice accumulation height $t_{acc}$ is calculated as follows (Brown and Määttänen, 2009):

$$t_{acc} = 3 + 4t \ , \tag{7}$$

$$t_{acc} = 7.6t^{0.64}, \tag{8}$$

where $t$ is level ice thickness.

**4.1.2 Sea ice risk modes and key structural parameters of oil platforms**

4.1.2.1 Extreme ice force may cause the overall structural failure

When the structural deformation under extreme ice force exceeds allowable deformation, it leads to a structural stiffness failure. When extreme ice force exceeds the ultimate bearing capacity of a structure, the structure is unstable.

4.1.2.2 Ice-induced vibration may cause structural fatigue failure

Structural fatigue damage is caused by a stress repeatedly applied at pipe joints, which are called hot spots. Structural hot spots fail after the significant stress ($S$) has been applied for the specified times ($N$). Long-term ice-induced structural fatigue damage may decrease the structural resistance and even cause the structural failure. In general, common ice loads have a higher probability and may cause a greater damage to a platform structure, whereas extreme ice loads have a lower probability and may cause a smaller effect on the platform.

4.1.2.3 Dynamic ice force may lead to the function failure of facility and affect personnel safety

The strong ice-induced vibration of a structure directly affects its upper facility. In such a vibration, the platform deck can be considered as a vibration table and cause whiplash on upper components. Especially, ice-induced vibration may lead to the direct damage to key functional facility or components without anti-vibration capability. Natural gas pipelines are distributed on the upper part of platforms. Due to the long-term ice-induced vibration, connecting parts of these pipelines may be weakened, thus causing natural gas leakage, pipeline breaking and even explosion. Ice-induced vibration may generate a slight impact on workers, decrease the comfort and efficiency, and affect health.

4.1.2.4 Sea ice accumulation may cause the damage to the upper facility or buildings

Sea ice accumulation increases the application area of ice force on marine structures as well as ice force itself. If sea ice climbs to a structure at a certain height from the ice surface along accumulated ice, it may cause the damage to the upper facility or buildings. If sea ice climbs to a dam, it destroys onshore buildings. If sea ice climbs to the underlying cable aisle of a platform, sea ice may crush the fence and affect the structural stability.

**4.2 Hazard index system**

**4.2.1 Determination of hazard indices based on the risk analysis of sea ice disasters**

Based on the relationships among sea ice parameters and their contributions to ice force, the key sea ice hazard indices were proposed (Table 1). Short-term sea ice hazard indices that play a key role in the failure of platforms include thickness, velocity, and strength. The long-term sea ice hazard indices related to the time and frequency of sea ice load include ice period and sea ice concentration.

**Table 1: Sea ice hazard indices.**

| Index types | Indices | Criteria |
|---|---|---|
| Short-term sea ice hazard indices | Ice thickness, ice velocity, and ice strength | Indices that play a key role in the failure of a platform |
| Long-term sea ice hazard indices | Ice period and sea ice concentration | Indices related to the time and frequency of sea ice load |

#### 4.2.2 Sea ice hazard indices

According to Technical Guidelines for Risk Assessment and Zoning of Sea Ice Disasters (State Oceanic Administration, 2016), various sea ice hazard indices were graded. The study area was divided into 21 regions and the values of the indices in each region were mainly determined based on the China Sea Ice Conditions and Application Regulations (Q/HSn 3000-2002). A return period of 100 years was selected in the subsequent analysis.

**Table 2: Grading of sea ice hazard indices.**

| Index codes | Indices | Index range | Extremely high hazard (5 points) | High hazard (4 points) | Medium hazard (3 points) | Low hazard (2 points) | Extremely low hazard (1 point) |
|---|---|---|---|---|---|---|---|
| $H_1$ | Designed ice thickness, cm | 8~41.7 | >35 | [35,25) | [25,10) | [10,5) | ≤5 |
| $H_2$ | Designed ice velocity, cm·s$^{-1}$ | 0.7~1.9 | >1.4 | [1.4,1.2) | [1.2,1.0) | [1.0,0.8) | ≤0.8 |
| $H_3$ | Designed ice strength, MPa | 1.88~2.37 | >2.2 | [2.2,2.1) | [2.1,2.02) | [2.02,1.9) | ≤1.9 |
| $H_4$ | Designed severe ice period, day | 30~149 | >35 | [35,25) | [25,10) | [10,5) | ≤5 |
| $H_5$ | Maximum ice concentration, % | 0~100 | >80 | [80,60) | [60,40) | [40,20) | ≤20 |

### 4.3 Vulnerability indices

#### 4.3.1 Determination of vulnerability indices based on the failure mode of jacket structures

According to typical sea ice disaster risk modes, the assessment indices of various structural failure modes ($M_1$, $M_2$, $M_3$, and $M_4$), structural vulnerability indices ($V_1$, $V_2$, $V_3$, and $V_4$) of sea ice disasters are proposed for the first time. The above assessment indices are calculated with structural parameters (Table 3) and structural factors are analyzed. The main failure modes of jacket structures include structure overturning by extreme ice force, structural fatigue damage caused by dynamic ice force, and the damage to the upper facility (including personnel) caused by dynamic ice force. Yue et al. (2007b) analyzed the static displacement of typical platforms under extreme ice force in the Liaodong Bay.

**Table 3: Sea ice disaster risk modes of oil platforms and corresponding vulnerability indices.**

| Risk modes | Structural performances | Assessment index by various structural failure modes | Structural vulnerability index | Structural parameters | Section |
|---|---|---|---|---|---|

| Structure overturning by the extreme ice force | Anti-overturning ability | Overturning index $M_1=V_1$ | Overturning index $V_1=K_n/KH$ | $H$, overall height of the structure; $K$, structural stiffness; $K_n$, coefficient | 4.3.1.1 |
|---|---|---|---|---|---|
| Structural fatigue damage caused by the dynamic ice force | Ice-induced vibration resistance capacity (displacement and strain) | the structural dynamic value corresponding to structural ice vibration fatigue $M_2=V_1*V_2$ | Overturning index $V_1=K_n/KH$ Dynamic index $V_2=\gamma_1*K_a$ | Above $\gamma_1$, dynamic amplification factor; $K_a$, coefficient of hot spots, $(0,1]$ | 4.3.1.2 |
| Facility damage caused by the dynamic ice force | Ice-induced vibration resistance capacity (acceleration) | The ice-induced vibration value $M_3$ $M_3=V_1*V_2*V_3*V_4^{0.5}$ | Overturning index $V_1=K_n/KH$ Dynamic index $V_2=\gamma_1*K_a$ Ice-induced vibration index $V_3=f_2$, Function index $V_4=K_b$ | Above $f$, first natural frequency for jacket structures $K_b$, the structural function coefficient, 1.5, 1.2, and 1.0 | 4.3.1.3 |
| Structural function failure caused by accumulation ice | Structural function | Damage to the upper facility of the structure caused by sea ice accumulation $M_4=V_4$ | Function index $V_4=K_b$ | $K_b$, the structural function coefficient, 1.5, 1.2, and 1.0 | 4.3.1.4 |

4.3.1.1 Structural overturning by extreme ice force (Mode 1) and structural overturning index

When extreme ice force exceeds the ultimate bearing capacity of a structure, the whole structure collapses. The overturning index $V_1$ is proposed below.

Based on functional descriptions of buildings under seismic loads, the damage of a structure under extreme ice load is provided in Table 4 (Ji and Yue, 2011).

**Table 4: Damages of marine structures under extreme ice loads.**

| Functional levels | I | II | III | IV | V |
|---|---|---|---|---|---|
| Damage states | Basically intact | Slight damage | Medium damage | Severe damage | Collapse |

| Relative structural deformation | $\Delta < H_{Str}/500$ | $H_{Str}/500 < \Delta < H_{Str}/250$ | $H_{Str}/250 < \Delta < H_{Str}/125$ | $H_{Str}/125 < \Delta < H_{Str}/50$ | $\Delta > H_{Str}/50$ |
|---|---|---|---|---|---|

Note: $\Delta = F/K$, where $F$ is ice force; $K$ is structural stiffness; $H_{Str}$ is the overall height of the structure.

When the ice force difference is not large (10~100 kN), the overturning index of the platform under extreme ice force is proposed as:

$$M_1 = V_1 = K_n/KH_{Str}, \tag{9}$$

where $H_{Str}$ is the overall height of the structure; $K$ is structural stiffness and set to be 10e7~10e9 for jacket platforms in the Liaodong Bay; $K_n$ is the coefficient related to the structural form (pile) and its values for the one-legged platforms and 4-legged platforms are respectively set to be 1 and 2 (Liu, 2007).

4.3.1.2 Structural fatigue damage caused by dynamic ice force (Mode 2) and structural dynamic index

Structural fatigue damage is caused by a stress repeatedly applied at hot spots of pipe nodes. For jacket structures, the stress applied at hot spots is usually linear with structural dynamic response ($\Delta d$), which is proportional to the static loading deformation ($\Delta = F/K$). The proportional coefficient is called the amplification factor $\gamma$ and directly related to structural natural frequency and ice force frequency. Yue et al. (2007b) analyzed the dynamic characteristics of anti-ice jacket platforms in the Bohai Sea. For the steady-state vibration of an upright structure, the amplification factor is:

$$\gamma = \frac{1}{\sqrt{(1-r^2)^2 + (2\xi r)^2}}. \tag{10}$$

For the random vibration of coned structures, the amplification factor is:

$$\gamma_1 = \frac{1}{5\sqrt{(1-r_1^2)^2 + (2\xi r_1)^2}}, \tag{11}$$

where both $r$ and $r_1$ are the frequency ratio, $r_1 = \omega/\omega_n = f/f_n$; $\omega(f)$ and $\omega_n(f_n)$ are respectively ice force frequency and structural natural frequency.

The structural dynamic index $V_2$ is calculated as:

$$V_2 = \gamma_1 * K_a, \tag{12}$$

where $\gamma_1$ is dynamic amplification factor and can be calculated with the measured data or frequency ratio ($\gamma_1 = f(\gamma_1,\xi)$) according to Eqs. (10) and (11); $K_a$ is the reinforcement coefficient of hot spots, namely, the ratio of the stress at the hot spot before reinforcement to that after reinforcement, and its range is (0,1]. Based on finite element analysis or measured data, in the study,

the values of $K_a$ are respectively selected as 0.5 for main platforms and satellite platforms and 1.0 for auxiliary platforms (Xu., 2014).

Considering the fatigue failure modes of jacket structures under dynamic ice force, with structural overturning index $V_1$ and dynamic index $V_2$, the structural dynamic value corresponding to structural ice vibration fatigue is expressed as:

$$M_2 = V_1 * V_2, \tag{13}$$

where $V_1$ is calculated according to Eq. (9).

4.3.1.3 Damage to the upper facility (including personnel) caused by dynamic ice force (Mode 3)

In general, the greater deck acceleration leads to the greater vibration amplitude of the facility. If a jacket structure can be simplified as a structure with a single degree of freedom (Yue et al., 2007b), deck vibration is similar to simple harmonic motion and its vibration displacement $D$, velocity $V$, and acceleration $A$ can be respectively expressed as:

$$D = \Delta st \times sin\,(\omega t + \varphi); \tag{14}$$

$$V = \Delta st \times \omega \times cos(\omega t + \varphi); \tag{15}$$

$$A = -\Delta st \times \omega^2 \times sin(\omega t + \varphi). \tag{16}$$

In Mode 2, vibration displacement $D$ corresponding to structural vibration index is the key factor to be considered. In the analysis of Mode 3, structural dynamic parameter, natural frequency $f$, should be carefully considered. The higher structural frequency means the greater acceleration. In addition, the structural function also directly affects the risk level. For example, there are many devices on oil production platforms. The design of manned platforms should consider personnel comfort since their risk level is relatively high. Unmanned platforms have a low risk level. In summary, structural ice vibration index $V_3$ and structural function index $V_4$ are proposed respectively based on natural frequency and structural function as follows:

$$V_3 = f^2, \tag{17}$$

where $f$ is the dominant ice vibration frequency of a platform, the first natural frequency for jacket structures.

$$V_4 = K_b, \tag{18}$$

where $K_b$ is the structural function coefficient and its values for manned central platforms, unmanned central platforms, and auxiliary function platforms such as the bollard are respectively set to be 1.5, 1.2, and 1.0 (Xu. 2014).

In Mode 3, the vibration and functions of a structure should be considered. The structural vulnerability indices to be considered include overturning index $V_1$, dynamic index $V_2$, ice-induced vibration index $V_3$, and function index $V_4$. The ice-induced vibration value $M_3$ is expressed as:

$$M_3 = V_1 * V_2 * V_3 * V_4^{0.5}, \tag{19}$$

where $V_1$, $V_2$, $V_3$, and $V_4$ are respectively calculated according to Eqs. (9), (12), (17), and (18).

The contribution of $V_4$ (Function index) is lower than other 3 structural vulnerability indexes ($V_1$ Overturning index, $V_2$ Dynamic index, $V_3$ Ice-induced vibration index), so 0.5 times of power was added to $V_4$ (Xu, 2014).

4.3.1.4 Damage to the upper facility of the structure caused by sea ice accumulation (Mode 4)

If sea ice climbs to the platform deck due to sea ice accumulation, it directly threatens the safety of facilities and personnel. Therefore, the vulnerability index mainly considered in Mode 4 is the functional index $V_4$.

$$M_4 = V_4 \ . \tag{20}$$

### 4.3.2. Vulnerability indices

According to the main distribution ranges of the parameters of jacket platforms in the Liaodong Bay, the above-mentioned structural vulnerability indices proposed based on the sea ice risk modes of oil platforms are graded into three levels: high, medium and low (Table 5).

**Table 5: Grading and assignment of structural vulnerability indices.**

| Index codes | Indices | Index range | High risk (5 points) | Medium risk (3 points) | Low risk (1 point) |
|---|---|---|---|---|---|
| $V_1$ | Overturning index | [4e-10,7e-9] | >2e-9 | [2e-9, 1e-9) | $\leq$ 1e-9 |
| $V_2$ | Dynamic index | [2,12] | >4 | [4,2) | $\leq$ 2 |
| $V_3$ | Ice-induced vibration index | [0.5,5] | >4 | [4,1.0) | $\leq$ 1.0 |
| $V_4$ | Function index | [1,1.5] | 1.5 | 1.2 | 1 |

### 4.4 Disaster resistance ability index

Emergency monitoring and sea ice management measures are the important factors to be considered in the assessment. The disaster resistance ability index $R_1$ is proposed (Zhang et al., 2012) and graded in three levels (Table 6).

**Table 6: Grading and assignment of disaster resistance ability index.**

| Index code | Index | Invalid I | Partially valid II | Valid III |
|---|---|---|---|---|
| $R_1$ | Disaster resistance ability index | 1.0 | (0.5, 1.0) | 0.5 |

### 4.5 Risk assessment method

Before the risk assessment of sea ice disasters, it is necessary to separately determine the index system, assessment models, and grading standards. The index system varies with the assessment model. The index systems are introduced separately according to the overall risk assessment method and the multi-mode risk assessment method below.

### 4.5.1 Overall risk assessment method

The weight coefficients of sea ice hazard indices were determined according to the importance of each index in the ice force calculation models (Table 7). The weights of structural vulnerability indices (Table 7) were determined based on the failure modes of structures and the probabilities of corresponding risks or accidents (Table 8).

**Table 7: Hierarchical structure and weights of sea ice risk assessment factors for the overall risk assessment.**

| Criteria layer | Index codes | Sub-criteria layer | Index codes | Weights | |
|---|---|---|---|---|---|
| Sea ice hazard indices | $H$ | Designed ice thickness | $W_1$ | 0.69 | |
| | | Designed ice velocity | $W_2$ | 0.02 | |
| | | Designed ice strength | $W_3$ | 0.06 | 1 |
| | | Maximum ice concentration | $W_4$ | 0.1 | |
| | | Designed severe ice period | $W_5$ | 0.13 | |
| Structural vulnerability indices | $V$ | Overturning index | $Q_1$ | 0.45 | |
| | | Overturning index | $Q_2$ | 0.39 | 1 |
| | | Ice-induced vibration index | $Q_3$ | 0.09 | |
| | | Functional index | $Q_4$ | 0.07 | |
| Structural resistance | $R$ | Structural resistance index | | 0.1 | |

**Table 8: Probability of platform failure modes and weight coefficient assignment.**

| Platform failure modes | Probability | Assessment index assignment | Weight coefficients | | | |
|---|---|---|---|---|---|---|
| | | | $V_1$ | $V_2$ | $V_3$ | $V_4$ |
| Structural overturning by extreme ice force | 6% | Overturning index assignment, $M_1=V_1$. | 0.06 | / | / | / |
| Dynamic ice force causes structural fatigue damage | 60% | Dynamic index assignment, $M_2=V_1*V_2$. | 0.3 | 0.3 | / | / |
| Dynamic ice force causes the damage to the upper facility (personnel) of structures | 30% | Ice-induced vibration index assignment, $M_3=V_1*V_2*V_3*V_4^{0.5}$. | 0.09 | 0.09 | 0.09 | 0.03 |
| Accumulated ice causes the damage to the upper facility of structures | 4% | Function index assignment, $M_4=V_4$. | / | / | / | 0.04 |
| Total | 100% | | 0.45 | 0.39 | 0.09 | 0.07 |

## 4.5.2 Multi-mode risk assessment method (multi-index synthetic risk assessment model)

Firstly, the weights of various risk modes were determined based on the failure modes of sea ice disasters and the probabilities of corresponding risks or accidents. Then, based on risk sources, risk mode assignments, and disaster resistance ability in various failure modes, the weights of the hazard indices, vulnerability indices, and disaster resistance ability indices were determined (Table 9).

**Table 9: Hierarchical structure and weights of sea ice risk assessment factors for multi-mode risk analysis.**

| Criteria layer | Index codes | Sub-criteria layer | Index codes | Weight coefficients | Key index layer | Index codes | Weight coefficients |
|---|---|---|---|---|---|---|---|
| Structural overturning by extreme ice force (0.06) | $R_1$ | Extreme ice force | $H_1$ | 0.20 | Ice thickness | $H_{1.1}$ | 0.20 |
| | | Anti-overturning capability of a platform | $V_1$ | 0.30 | Structural overturning index | $V_{1.1}$ | 0.30 |
| | | Disaster resistance ability | $R_1$ | 1.00 | Disaster resistance ability index | $R_{1.1}$ | 1.00 |
| Damage to the main structure caused by ice-induced vibration (0.60) | $R_2$ | Dynamic ice force and its influencing scope (temporal and spatial distributions) | $H_2$ | 1.00 | Ice thickness | $H_{2.1}$ | 0.8 |
| | | | | | Ice period | $H_{2.2}$ | 0.2 |
| | | Anti-overturning capability of a platform | $V_2$ | 0.60 | Structural overturning index | $V_{2.1}$ | 0.30 |
| | | | | | Dynamic index | $V_{2.2}$ | 0.30 |
| | | Disaster resistance ability | $R_1$ | 1.00 | Disaster resistance ability index | $R_{2.1}$ | 1.00 |
| Damage to the upper facility caused by ice-induced vibration (0.30) | $R_3$ | Dynamic ice force and its influencing scope | $H_3$ | 0.50 | Ice thickness | $H_{3.1}$ | 0.30 |
| | | | | | Ice concentration | $H_{3.2}$ | 0.10 |
| | | | | | Ice period | $H_{3.3}$ | 0.10 |
| | | Anti-overturning capability and function of a platform | $V_3$ | 0.60 | Structural overturning index | $V_{3.1}$ | 0.18 |
| | | | | | Structural dynamic index | $V_{3.2}$ | 0.18 |
| | | | | | Ice-induced vibration index | $V_{3.3}$ | 0.18 |
| | | | | | Function index | $V_{3.4}$ | 0.06 |
| | | Disaster resistance ability | $R_1$ | 1.00 | Disaster resistance ability index | $R_{3.1}$ | 1.00 |
| Damage to the facility caused | $R_4$ | Ice accumulation | $H_4$ | 0.50 | Ice thickness | $H_{4.1}$ | 0.40 |
| | | | | | Ice concentration | $H_{4.2}$ | 0.1 |
| | | Function of a platform | $V_4$ | 0.08 | Function | $V_{4.1}$ | 0.08 |

| by accumulation ice (0.04) | Disaster resistance ability | $R_1$ | 1.00 | Disaster resistance ability index | $R_{4.1}$ | 1.00 |
|---|---|---|---|---|---|---|

## 4.6 Assessment calculation method and grading criteria

According to Eqs. (1) to (3) in Section 3.2, the risk was calculated with the overall risk analysis method and then graded into 4 levels. The criteria and results of the risk assessment of sea ice disasters on oil platforms are proposed in this study (Table 10).

Table 10: Assessment criteria of the risks of sea ice disasters on the oil platforms in the Bohai Sea.

| Risk index | (12, 25] | (9, 12] | (6, 9] | [0.5, 6] |
|---|---|---|---|---|
| Levels | Severe risk | Moderate risk | Mild risk | Low risk |

## 5 Case analysis

### 5.1 Parameters

Taking 10 jacket platforms with different functions in the three regions of Liaodong Bay (JZ20-2, JZ21-1, and JZ9-3) as the cases, sea ice risks were calculated with the above assessment methods. The vulnerability index was determined according to the locations of the three regions and corresponding sea ice parameters (Table 11). The designed and assigned values of the vulnerability indices of the 10 platforms were determined by the basic forms, functions and dynamic parameters of the platforms (Table 12).

Table 11: Designed and assigned values of sea ice hazard indices in case analysis

| Indices | Sea regions | Sea Region 20-2 | Sea Region 21-1 | Sea Region 9-3 |
|---|---|---|---|---|
| Designed ice thickness ($H_1$) | Designed values/cm | 41.7 | 40.4 | 36.8 |
| | Assigned values | 5 | 5 | 5 |
| Designed ice velocity ($H_2$) | Designed values /cm·s$^{-1}$ | 1.9 | 1.8 | 1.4 |
| | Assigned values | 5 | 5 | 4 |
| Designed ice strength ($H_3$) | Designed values /Mpa | 2.37 | 2.16 | 2.33 |
| | Assigned values | 5 | 5 | 5 |
| Designed severe ice period ($H_4$) | Designed values /day | 85 | 53 | 72 |

| | | | |
|---|---|---|---|
| Assigned values | 5 | 5 | 5 |
| | Designed values /% | almost 100 | almost 100 | 80 |
| Maximum ice concentration ($H_5$) | Assigned hazard values | 5 | 5 | 5 |

**Table 12: Designed and assigned values of structural vulnerability indices in the case analysis.**

| Platforms | | JZ20-2 A | JZ20-2 B | JZ20-2 C | JZ20-2 D | JZ21-1 E | JZ9-3 F | JZ9-3 G | JZ9-3 H | JZ9-3 I | JZ9-3 J |
|---|---|---|---|---|---|---|---|---|---|---|---|
| Quantity of legs | | 4 | 3 | 4 | 1 | 4 | 4 | 1 | 4 | 1 | 1 |
| Leg forms | (Cone or Cylinder) | cone | cone | cone | cone | cone | cone | cylinder | cone | cylinder | cone |
| Platform functions | (Oil recovery/auxiliary function) | Oil recovery | Oil recovery | Oil recovery | Oil recovery | Oil recovery | Auxiliary compressor | Auxiliary mooring pile | Oil recovery | Auxiliary mooring pile | Oil recovery |
| Manned/unmanned | | Manned | unmanned | Manned | Manned | unmanned | unmanned | unmanned | Manned | unmanned | unmanned |
| Leg coefficients | $k_n$ | 1.5 | 2 | 1.5 | 1 | 1.5 | 1.5 | 1 | 1.5 | 1 | 1.2 |
| Static stiffness | $K$ | 2.00E+08 | 6.40E+07 | 9.30E+07 | 6.10E+07 | 9.00E+07 | 1.20E+08 | 1.30E+08 | 9.00E+07 | 5.40E+07 | 2.10E+07 |
| Water depth | $H$ | 15.6 | 16.5 | 16.5 | 13.5 | 15.6 | 9.5 | 10 | 9 | 9 | 9 |
| Amplification coefficient | $\gamma$ | 4.17 | 4.17 | 4.17 | 4.17 | 4.17 | 6 | 12 | 4.17 | 4.17 | 15 |
| Ice-breaking coefficient | $K_a$ | 0.5 | 0.5 | 0.5 | 0.5 | 0.5 | 1 | 1 | 0.5 | 1 | 0.5 |
| Natural frequency | $f$ | 0.87 | 1.36 | 1.41 | 1 | 1.1 | 2.06 | 2.32 | 1.1 | 6.4 | 0.84 |
| Function coefficient | $K_b$ | 1.5 | 1.2 | 1.5 | 1.5 | 1 | 1.2 | 1 | 1.5 | 1 | 1.2 |
| Anti-overturning index | $V_1=k_n/(KH)$ | 4.80E-10 | 1.90E-09 | 9.80E-10 | 1.20E-09 | 1.10E-09 | 1.30E-09 | 7.60E-10 | 1.90E-09 | 2.10E-09 | 6.50E-09 |
| | Assigned values | 1 | 3 | 1 | 3 | 3 | 3 | 1 | 3 | 5 | 5 |
| Dynamic index | $V_2=K_a*\gamma$ | 2.08 | 2.08 | 2.08 | 2.08 | 2.08 | 6 | 12 | 2.08 | 4.17 | 7.5 |
| | Assigned values | 3 | 3 | 3 | 3 | 3 | 5 | 5 | 3 | 5 | 5 |
| Ice-induced vibration index | $V_3=f^2$ | 0.76 | 1.85 | 1.99 | 1 | 1.21 | 4.24 | 5.38 | 1.21 | 40.96 | 0.71 |
| | Assigned values | 1 | 3 | 3 | 1 | 1 | 5 | 5 | 3 | 5 | 1 |
| Function index | $V_4=k_b$ | 1.5 | 1.2 | 1.5 | 1.5 | 1 | 1.2 | 1 | 1.5 | 1 | 1.2 |
| | Assigned values | 5 | 3 | 5 | 5 | 1 | 3 | 1 | 5 | 1 | 3 |

## 5.2 Sea ice risk assessment and grading

With the overall risk analysis method described in Section 4.4.1, the sea ice hazard ($H$), structural vulnerability ($V$) and disaster resistance ability ($R$) were determined and then the overall risk $I_e$ was calculated according to Eq. (1) and Table 7. Then, the calculation results of four sea ice risks $I_{s,i}$ (i=1,2,3,4) were determined. According to Eqs. (2) and (3), with the synthetic index method, the multi-mode risk analysis results $I_s$ and maximum risk values $I_{s,max}$ were calculated.

**Table 13: Sea ice risk assessment analysis and risk grading results in the case analysis.**

| Platforms | H | V | R | Ie=HVR | $I_{s,1}$ | $I_{s,2}$ | $I_{s,3}$ | $I_{s,4}$ | $I_{s,max}$ | $I_{s,c}$ |
|---|---|---|---|---|---|---|---|---|---|---|
| JZ20-2 A | 5 | 2.06 | 0.5 | 5.15 | 2.5 | 5 | 5 | 12.5 | 12.5 | 5.15 |
| JZ20-2 B | 5 | 3 | 0.5 | 7.5 | 7.5 | 7.5 | 7.5 | 7.5 | 7.5 | 7.5 |
| JZ20-2 C | 5 | 2.24 | 0.5 | 5.6 | 2.5 | 5 | 6.5 | 12.5 | 12.5 | 5.6 |
| JZ20-2 D | 5 | 2.96 | 0.5 | 7.4 | 7.5 | 7.5 | 6.5 | 12.5 | 12.5 | 7.4 |
| JZ21-1 E | 5 | 2.68 | 0.5 | 6.7 | 7.5 | 7.5 | 5.5 | 2.5 | 7.5 | 6.7 |
| JZ9-3 F | 4.98 | 3.96 | 0.5 | 9.86 | 7.5 | 10 | 11 | 7.5 | 10.5 | 9.9 |
| JZ9-3 G | 4.98 | 2.92 | 0.5 | 7.27 | 2.5 | 7.5 | 8.5 | 2.5 | 8.5 | 7.3 |
| JZ9-3 H | 4.98 | 3.14 | 0.5 | 7.82 | 7.5 | 7.5 | 8 | 12.5 | 12.5 | 7.85 |
| JZ9-3 I | 4.98 | 4.72 | 0.5 | 11.8 | 12.5 | 12.5 | 12 | 2.5 | 12.5 | 11.8 |
| JZ9-3 J | 4.98 | 4.5 | 0.5 | 11.2 | 12.5 | 12.5 | 9 | 7.5 | 12.5 | 11.25 |

Overall risk analysis: H, V, R, Ie=HVR. Multi-mode risk analysis: $I_{s,1}$, $I_{s,2}$, $I_{s,3}$, $I_{s,4}$, $I_{s,max}$, $I_{s,c}$.

Notes: ■ indicates severe risk; ■ indicates moderate risk; ■ indicates mild risk; ■ indicates low risk.

## 5.3 Analysis results

The three risk calculation results ($I_e$, $I_s$, and $I_{s,max}$) were analyzed (Fig. 5). In the calculation results obtained by the synthetic index method, the overall risk analysis results $I_e$ were basically the same to the multi-mode risk analysis results $I_s$ and the risk grading results were the same because the theoretical basis for establishing the index system and the weights of the secondary indices adopted in the two methods were the same. The risk mode with the higher weight (such as Mode 2) dominated the multi-mode risk analysis results ($I_s$) obtained with the synthetic index method. When $I_{s,max}$ was significantly different from $I_e$ and $I_s$, the risk values of most of the risk modes (such as Mode 4) with lower weights were higher, such as the risks of Platforms A, C, D, and H.

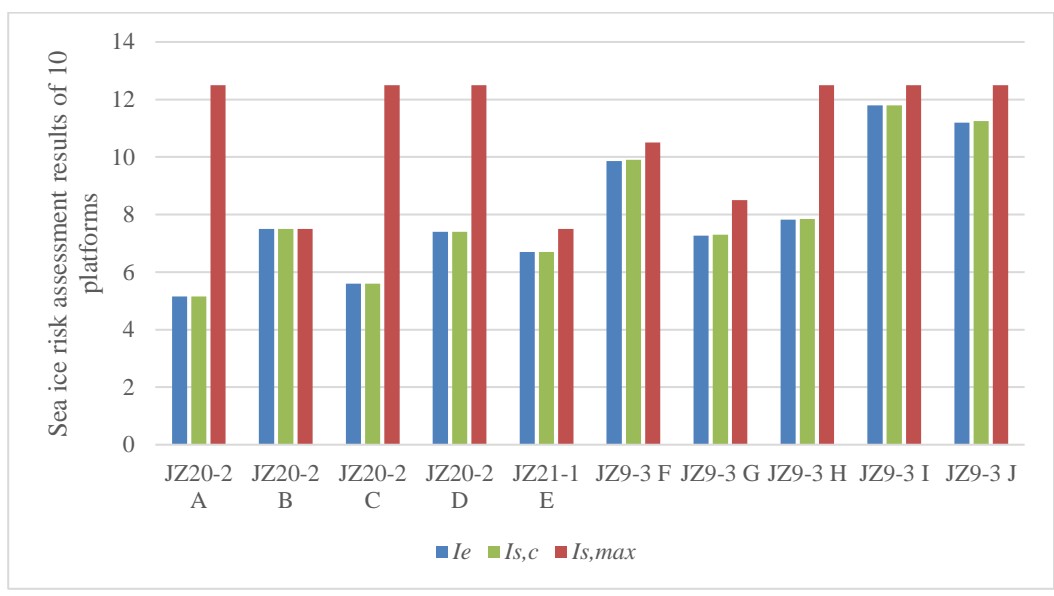

**Figure 4: Comparison of the three risk calculation results ($I_e$, $I_s$, and $I_{s,max}$).**

The overall risk analysis results $I_e$ and the multi-mode risk analysis results $I_s$ indicate that two main reasons are responsible for the higher risk level. Firstly, the steady-state vibration may occur on the structures and the dynamic larger amplification factor $\gamma$ leads to the higher dynamic index $V_2$. Therefore, the structural fatigue failure related to ice-induced vibration caused by dynamic ice force (Mode 2) occurs on some platforms, such as Platforms F and J. Secondly, due to the high structural fundamental frequency, the structural ice-induced vibration index $V_3$ is large and the facility function failure caused by ice-induced vibration acceleration (Mode 3) occurs. For example, Platform I has a fundamental frequency of 6.4 Hz, which is significantly higher than the fundamental frequency of common jacket structures in the Liaodong Bay (0.5~2 Hz).

## 6 Conclusions

In the study, the risk assessment method of sea ice disasters was developed for jacket platforms in ice-covered sea areas in the Liaodong Bay of Bohai Sea. The sea ice risk index system considering sea ice hazard, structural vulnerability and disaster resistance ability was established. In addition, based on the synthetic index method, sea ice disaster assessment methods were constructed, including the overall risk assessment method and multi-mode risk assessment method. The above key indices were determined based on the formation mechanism of sea ice disasters. The weights of these indices were recommended based on the previously recorded sea ice disaster cases.

This paper focuses on the structural risk induced by level ice. The assessment method is also applicable to rafted ice since it has the similar ice-structure interaction process with level ice. In addition, the differences in ice parameters between rafted ice and level ice should be considered, such as the range of ice thickness, the value of ice strength, and the their weight for the

risk value under different risk modes. The values of these indices were determined based on the ice conditions and parameters of jacket platforms in the Liaodong Bay, so the applicability of these values in other sea areas needs to be further verified. The assessment system is a qualitative description of risk and can be applied in structural optimization during the design phase and real-time risk level assessment during the operation phase. In the future, we will make the detailed analysis based on the

5 preliminary results obtained with the risk assessment method in order to provide more efficient and accurate assessment results.

### Acknowledgement

This study is financially supported by The National Key Research and Development Program of China (Grant No. 2017YFQ0604902), Public Science and Technology Research Funds Projects of Ocean (Grant No. 201505019), and National Natural Science Foundation of China (Grant No. 41676087).

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
