# Peer review of "Risk assessment of sea ice disasters on fixed jacket platforms in the Liaodong Bay"

_Natural Hazards and Earth System Sciences, 2018_

## Referee Comment (RC1) · Anonymous Referee #1 · 22 Feb 2019

The paper Risk assessment of sea ice disasters on fixed jacket platforms in the Liaodong Bay focused on how to quantify the risk potentially caused by sea ices to the jacket platforms in the three sea areas in the Liaodong Bay. This work analyzed in great details the formation mechanism of sea ice hazards including their sources and modes. Two calculation methods based on the so-called overall risk assessment and multi-mode risk assessment were proposed and were applied to 10 jacket platforms located in three sea areas of Liaodong Bay. This work has great practical significance and can raise wide interests from the corresponding offshore engineering fields.

Before consideration of this paper for publication, some issues must be further explained/improved by the authors. One major issue is that the authors have not included corresponding citations in the text, though it is presumed that some parameter num-

bers are suggested by traditional books. It is thus difficult for readers to figure out how and where those parameters or indices values are calculated or selected. It is also difficult to judge which parts were originally proposed by the authors. Another major issue is that only the literature from Chinese researchers/engineers is mentioned. How the international community is treating the similar problems? What are the existing methods used in practices to evaluate the risk indices? This is not clear yet to readers.

Many small issues for the authors' reference: Page 2, Line 6: It should be ';' after Li et al., 2018. Page 2, Line 12: It should be Guo et al. (2018). Page 4: In Figure 2 it is better mark (a)-(e) on the photo as well. Page 5, Line 4: 'According to the synthetic index method, ...', reference? Page 5, Line 17: Add (Eq. (2)) after 'the overall risk Ie is calculated'. Pages 5-6: The position of Eqs. (1)-(3) should be adjusted to the corresponding text places. Page 5, Lines 20-21: Note that parameter name should be italic. Page 6: Please provide references to Eqs. (4) and (5). Page 7, Line 2: The parameter should be italic. Page 7, Lines 5-16: References should be provided on how to calculate force, including the source of Eq. (6). Page 8: What are the differences between Eqs. (7) and (8)? Reference? Page 8, Line 10: What are the hot spots? Page 8, Line 11: Note parameter name should be italic. Page 9, Line 9: 'According to Technical Guidelines for Risk Assessment and Zoning of Sea Ice Disasters, ...', reference? Page 9, Table 2: The last column, delete 'cm'. Page 10, Line 6: It should be Yue et al. (2007b). Page 10, Table 3: The last column, section numbers are not right. Page 11: Add reference to Eq. (9). Page 11, Lines 7-8: Note parameter names should be italic. Page 11, Line 9: Correct the double ')' after 2007b. Page 11, Lines 16-17: Reference for Eq. (12)? Parameter name italic? What is frequency ratio? Page 12, Lines 3-7: Parameter names italic? What is $\Delta\_st$? Page 12, Lines 13-14: f2 or f? What does fundamental frequency mean? Page 12, Lines 15-19: Reference? Page 12: Please explain where 0.5 comes in Eq. (19). Reference? Page 13, Line 2: Please provide the reference. Page 13: The numbering for Table 7 and Table 8 should be exchanged since the latter was mentioned first in the text. Page 13: In the original Table 7, weight coefficients should not be bold. Page 14: In the original Table 7, last

row, the position of 100% is not right. Page 14: In the original Table 8, could the authors explain why the weights are not added to 1? Page 15: The section name 4.6 should be bold. Page 15: Reference for Table 10? Page 16: Maybe the authors consider rotating Tables 11 and 12 90 degrees so that the readers can more easily read? Page 18: In Table 13 the parameter name should be italic. Page 19, Line 5: The label for the amplification factor is wrong. Pages 20-21: Reference style is not consistent.

---

## Referee Comment (RC2) · Shunying Ji (Referee) · 23 Feb 2019

The manuscript suggests a risk assessment for fixed jacket platforms in icy waters. The overall criterion is based on valuable full-scale data from different platforms located in the Bohai Sea. Besides measured structure responses, this work considers real physical process including ice-structure interaction and structure failure modes, which yields to several risk modes and levels. In the risk analysis, the authors give specific details describing different structure failure potentials. This work has a great value for offshore structure design in ice-covered waters. In believe this manuscript is subjected to a minor revision.

Normally, the ice condition is relative weak in the Bohai Bay. In my knowledge, the ice

type includes level ice, rafted ice and small ridges. My only concern is that for a given ice thickness if the ice type may influence the risk assessment.

Furthermore, some language corrections own to be dealt with. Line 9, page 1, the risk assessment should be a risk assessment. Line 14, page, facility should be facilities. Line 22, page 1, occurred should be occurs. Line 13, page 2, parameters of should be parameters, i.e.,. Line 17, page 2, application should be engineering. Line 23, page 1, respectively should be individually. Line 24, page 2, good condition should be healthy condition. Line 27, page 2, iced should be icy. Line 2, page 3, more serious than should be heavier than. Line 10, page 2, forms should be categories. Line 14, page 2, form should be category. Line 14, page 5, "was established with the risk calculation model to" to "with the risk model was established to". Line 10, page 6, delete "the interactions between". Line 16, page 6, "contacts with" should be "contacts". Line 5, page 7, "it will generate" to "it generates". Line 10, page 8, "will fail" to "fails". Line 20, page 8, "loosened" to "weakened". Line 10, page 10, "will collapse" to "collapses". Line 8, page 12, "additionally" to "carefully". Line 18, page 13, "sea ice disasters" to "structures".

---

## Author Comment (AC1) · 29 Mar 2019

1ïïjŇ (1) comments from Referees One major issue is that the authors have not included corresponding citations in the text, though it is presumed that some parameter numbers are suggested by traditional books. It is thus difficult for readers to figure out how and where those parameters or indices values are calculated or selected. It is also difficult to judge which parts were originally proposed by the authors.

(2) author's response Corresponding citations was added in the revision. And the parameter numbers were highlight which were proposed for the first time, i.e., Assessment index by various structural failure modes (M1(Eq. (9)),M2(Eq. (13)),M3(Eq. (19)),M4(Eq. (20))), structural vulnerability indices (V1(Eq. (9)),V2(Eq. (12)),V3(Eq.

(17)),V4(Eq. (18)), , Eq. (19)) and Assessment criteria (Table 10).

(3) author's changes in manuscript. Corresponding citations was added in the revision. And the parameter numbers were highlight which were proposed for the first time, i.e., ïĄň Assessment index by various structural failure modes: M1(Eq. (9)),M2(Eq. (13)),M3(Eq. (19)),M4(Eq. (20)), ïĄň structural vulnerability indices: V1(Eq. (9)),V2(Eq. (12)),V3(Eq. (17)),V4(Eq. (18)), , Eq. (19) and ïĄň Section 4.6: Assessment criteria (Table 10)

2ïijŇ (1) comments from Referees Another major issue is that only the literature from Chinese researchers/engineers is mentioned. How the international community is treating the similar problems? What are the existing methods used in practices to evaluate the risk indices? This is not clear yet to readers.

(2) author's response Risk assessment studies of engineering structures under environmental loads aremainly focused on large-span bridges, houses and other buildings under wind and seismic loads, and corresponding vulnerability curves were obtained. While there are little conclusions and applications of risk assessment studies of sea ice loads on marine platforms, no risk assessment indexes and method. The fundamental studies are mainly focused on ice force calculation methods, structural failure mode analysis, ice force resistance of engineering structures, and fatigue life calculation. The above work would be the basic theory for the sea ice risk assessment indicator system and assessment method of marine structures.

(3) author's changes in manuscript. The above contents were added in section "INTRODUCTION", Page 2, line 18-24 Corresponding citations was added in the revision.

3ïijŇ (1) comments from Referees Many small issues for the authors' reference: Page 2, Line 6: It should be ';' after Li et al., 2018. Page 2, Line 12: It should be Guo et al. (2018). Page 4: In Figure 2 it is better mark (a)-(e) on the photo as well. Page 5, Line 4: 'According to the synthetic index method, : : :', reference? Page 5, Line 17: Add (Eq. (2)) after 'the overall risk Ie is calculated'. Pages 5-6: The position of Eqs. (1)-(3)

should be adjusted to the corresponding text places. Page 5, Lines 20-21: Note that parameter name should be italic. Page 6: Please provide references to Eqs. (4) and (5). Page 7, Line 2: The parameter should be italic. Page 7, Lines 5-16: References should be provided on how to calculate force, including the source of Eq. (6). Page 8: What are the differences between Eqs. (7) and (8)? Reference? Page 8, Line 10: What are the hot spots? Page 8, Line 11: Note parameter name should be italic. Page 9, Line 9: 'According to Technical Guidelines for Risk Assessment and Zoning of Sea Ice Disasters, : : :', reference? Page 9, Table 2: The last column, delete 'cm'. Page 10, Line 6: It should be Yue et al. (2007b). Page 10, Table 3: The last column, section numbers are not right. Page 11: Add reference to Eq. (9). Page 11, Lines 7-8: Note parameter names should be italic. Page 11, Line 9: Correct the double ')' after 2007b. Page 11, Lines 16-17: Reference for Eq. (12)? èǦłåőŽáźĽParameter name italic? What is frequency ratio? Page 12, Lines 3-7: Parameter names italic? What is __st? Page 12, Lines 13-14: f2 or f? What does fundamental frequency mean? Page 12, Lines 15-19: Reference? èǦłåőŽáźĽ Page 12: Please explain where 0.5 comes in Eq. (19). Reference? èǦłåőŽáźĽ Page 13, Line 2: Please provide the reference. Page 13: The numbering for Table 7 and Table 8 should be exchanged since the latter was mentioned first in the text. Page 13: In the original Table 7, weight coefficients should not be bold. Page 14: In the original Table 7, last row, the position of 100% is not right. Page 14: In the original Table 8, could the authors explain why the weights are not added to 1? Page 15: The section name 4.6 should be bold. Page 15: Reference for Table 10? èǦłåőŽáźĽ Page 16: Maybe the authors consider rotating Tables 11 and 12 90 degrees so that the readers can more easily read? Page 18: In Table 13 the parameter name should be italic. Page 19, Line 5: The label for the amplification factor is wrong. Pages 20-21: Reference style is not consistent.

(2) author's response A: Every issues has been revised, except the following: a. I don't understand the comments on "Page 5, Line 17: Add (Eq. (2)) after 'the overall risk Ie is calculated'. " and "Page 12, Lines 3-7: What is __st? " b. The parameter n which were proposed for the first time were indicated as comment 1, including

the following list items. Page 11: Add reference to Eq. (9). Page 11, Lines 16-17: Reference for Eq. (12)? Page 12, Lines 15-19: Reference? Page 15: Reference for Table 10? c. Page 12: Please explain where 0.5 comes in Eq. (19). The ice-induced vibration value M3 is the third assessment index by various structural failure modes, which is expressed as: M3=V1*V2*V3*V40.5, (19) where V1, V2, V3, and V4 are respectively calculated. Because the contribution of V4 (Function index) is lower than other 3 Structural vulnerability indexes (V1 Overturning index,V2 Dynamic index,V3 Ice-induced vibration index), so the author added the 0.5 times power on V4. (3) author's changes in manuscript. The draft was revised following the comments. And the explanation for Eq. (19) was added on in page 13, Line 10-11.

Please also note the supplement to this comment:
https://www.nat-hazards-earth-syst-sci-discuss.net/nhess-2018-280/nhess-2018-280-AC1-supplement.pdf

---

## Author Comment (AC2) · 29 Mar 2019

1ïïjŇ (1) comments from Referees Normally, the ice condition is relative weak in the Bohai Bay. In my knowledge, the ice type includes level ice, rafted ice and small ridges. My only concern is that for a given ice thickness if the ice type may influence the risk assessment.

(2) author's response This paper focus on the structural risk induced by level ice, based on the formation mechanisms of sea ice disasters, which is the level ice failure process while interacting on structures and level ice force on structures. There are mainly three mode of ice force, corresponding to different limit rules: (a)limit stress (b) limit momentum (c)limit forceãĂĆ

[Figure]

For the Bohai Sea, the level ice and rafted ice would follow the limit stress rules. While the ridge would mainly follow limit momentum rule. Since the mechanisms of rafted ice and structure interaction process was similar with level ice, the authors think the method could also be used on rafted ice, while the variance of ice property parameters value should be paid attention.

(3) author's changes in manuscript. Some words are added in the section "CONCLU-SION" page 20 Line 6-9. "This paper focus on the structural risk induced by level ice, based on the formation mechanisms of sea ice disasters, which is the level ice failure process while interacting on structures and level ice force on structures. The method could also be used on rafted ice because of the similar ice-structure interaction process with level ice. While the variance of ice property parameters value should be paid attention."

2ïïjŇ (1) comments from Referees Furthermore, some language corrections own to be dealt with. Line 9, page 1, the risk assessment should be a risk assessment. Line 14, page1, facility should be facilities. Line 22, page 1, occurred should be occurs. Line 13, page 2, parameters of should be parameters, i.e.,. Line 17, page 2, application should be engineering. Line 23, page 1, respectively should be individually. Line 24, page 2, good condition should be healthy condition. Line 27, page 2, iced should be icy. Line 2, page 3, more serious than should be heavier than. Line 10, page 2, forms should be categories. Line 14, page 2, form should be category. Line 14, page 5, "was established with the risk calculation model to" to "with the risk model was established to". Line 10, page 6, delete "the interactions between". Line 16, page 6, "contacts with" should be "contacts". Line 5, page 7, "it will generate" to "it generates". Line 10, page 8, "will fail" to "fails". Line 20, page 8, "loosened" to "weakened". Line 10, page 10, "will collapse" to "collapses". Line 8, page 12, "additionally" to "carefully". Line 18, page 13, "sea ice disasters" to "structures".

(2) author's response All the listed errors have been revised, and English of the whole article has been revised.

(3) author's changes in manuscript. All the listed errors have been revised, and English of the whole article has been revised.

Please also note the supplement to this comment:
https://www.nat-hazards-earth-syst-sci-discuss.net/nhess-2018-280/nhess-2018-280-AC2-supplement.pdf

―――――――――――――――――

[Figure]

[Figure]

Fig. Three limit rules for calculating extreme ice force: (a)limit stress (b) limit momentum (c)limit force

**Fig. 1.**

---

## Author Response (AR2)

**Response to editor and reviewer**

**Editor: Heidi Kreibich**

• Comment 1:

(a) NOVELTY OF YOUR STUDY. Your research on sea ice disasters is interesting and very relevant from a practical point of few. Please try to strengthen the description of its scientific novelty. This needs to be done both at the beginning so we understand, but also in discussion, telling us 'why should someone outside of your study area be interested in the results'. If you were to explain the results of your case study to someone in another country, what would they gain from your study? What is novel and what might they learn?

Answer:

The following words are added in the INTROUDCTION.

"The optimization selection and safety assessment of engineering structures generally involve the long-term simulation calculation and require a certain calculation basis. If an assessment index system is established based on the risk assessment theory, the safety level of various structures and the risk level of structural operation can be quickly assessed. Therefore, a structural risk assessment index system can be used as a basis for the more efficient and accurate structural safety analysis."

• Comment 2:

(b) BROADER CONTEXT OF YOUR STUDY. You do not relate your work to the broader international literature of what others have done. We need to understand this broader context and what others have done.

Answer: Four literatures were added in the first paragraph of introduction.

Mangalathu, S., Jong-Su, J., Padgett J. E., Reginald, D.: Performance-based grouping methods of bridge classes for regional seismic risk assessment: Application of ANOVA, ANCOVA, and non-parametric approaches. Earthquake Engineering & Structural Dynamics, 2017, 46(1).

Wu Q.Y., Zhu H. P., Fan J.: Performance-based seismic financial risk assessment of, reinforced concrete frame structures[J]. Journal of Central South University, 2012, 19(5):1425-1436.

Tromans, P. S., Van d G. J. W.: Substantiated Risk Assessment of Jacket Structure. Journal of Waterway, Port, Coastal, and Ocean Engineering, 1994, 120(6):535-556.

Melani, A., Khare R., Dhakal R.P., Mander, J.B.: Seismic risk assessment of low rise RC frame structur. Structures, 5:13-22.

• Comment 3:

(e) ENGLISH. Although your manuscript will undergo a copy editing at the final stage, there are sentences in your manuscript which one cannot follow well due to the issues of English. I suggest an improvement of the English language.

A: The English of this paper has been revised.

• Comment 4:

(f) GENERAL TECHNICAL FORMATTING. I encourage you to examine carefully the NHESS author guidelines for formatting available online at: https://www.natural-hazards-andearth-system-sciences.net/for\_authors/manuscript\_preparation.html.

A: Format modification including title, author, graph and table.

**#Revewer 3**

• Comment 1:

Grammar needs to be corrected throughout the paper.

Answer: The English of this paper has been revised.

• Comment 2:

Technical discussion needs to be worded in state-of-the-art terms. For example: "When the overall deformation of a structure exceeds its allowable deformation under extreme ice force, structural stiffness failure occurs." While I follow the meaning of the sentence, it is worded awkwardly and is not consistent with the current state of practice in structural engineering. Another example would be "ice pile climbing." What is meant by this?

Answer:

(1)" When the overall deformation of a structure exceeds its allowable deformation under extreme ice force, structural stiffness failure occurs" was revised to" When the structural deformation under extreme ice force exceeds allowable deformation, it leads to a structural stiffness failure.".

(2)"ice pile climbing" was revised to " ice pile climbing".

**• Comment 3:**

The authors should explain how the index will be used in practice. It appears to result in a relative description of risk and not an absolute measure. How will that description be used? A: "The assessment system is a qualitative description of risk and can be applied in structural optimization during the design phase and real-time risk level assessment during the operation phase. In the future, we will make the detailed analysis based on the preliminary results obtained with the risk assessment method in order to provide more efficient and accurate assessment result."

The above words were added in the conclusion.

**• Comment 4:**

the discussion of dynamics is probably more detailed than is need for this paper. Explanation of when higher modes are considered is probably all that is required.

A:"4.1.1.2 Dynamic ice force" was trimmed. Figure 4 was deleted.

**• Comment 5:**

The 0.5 exponent on V4 appears to be arbitrary. Why was that value chosen? Does it have significance?

Answer: the structural function also directly affects the risk level. For example, there are many devices on oil production platforms. The design of manned platforms should pay attention to personnel comfort and their risk is relatively high. Unmanned platforms have a low risk. Kb is the structural function coefficient and its values for manned central platforms, unmanned central platforms, and auxiliary function platforms such as the bollard are

respectively set to be 1.5, 1.2, and 1.0.

The ice-induced vibration value M3 is expressed as:M3=V1\*V2\*V3\*V40.5.

By compare contribution of four parameters, which are overturning index V1, dynamic indexV2, ice-induced vibration indexV3, and function indexV4= Kb), The value of V4 made smaller contribution to risk results, so the 0.5 exponent of proposed. This conclusion has been validated by experts, and the conclusion was published in postdoctoral research report" Xu, N. Research on Critical Issues of Sea-ice Disaster Risk Assessment and Prevention Strategy. 2014".

The above literature was added

**• Comment 6:**

Line 12: It was stated that a 100-year return interval was chosen because the structure has a 100-year design life. These two design criterion do not correlate.

**This literature was added in this paper.**

A: "Since the designed life of the oil platform in Bohai Sea was 100 years, a return period of 100 years was selected in the subsequent analysis." was revised to "A return period of 100 years was selected in the subsequent analysis"

**• Comment 7:**

For Table 3, the division between the rows needs to be better defined. It is not clear where one row start and stops, vertically.

Answer: Black lines were added between various rows.

**• Comment 8:**

It is stated in the text that Table 4 summarizes relative structural deformation for buildings. How do we know these drift limits are applicable to offshore jacketed structures?

Answer: one literature was added. This conclusion was published in "Ji,S.Y. and Yue, Q.J.: Numerical model on engineering sea ice and its application. Science Press. 327-328. 2011"

---

## Author Response (AR3)

**Response to editor and reviewer**

**Editor: Heidi Kreibich**

- **I advise you to take the chance to check your manuscript again and to make sure, that all equations, expressions and definitions are correct and to further improve as much as possible.**

Answer:

The manuscript has been checked including all of the equations, expressions and definitions.

**Anonymous Referee #1**

This research paper has developed an efficient risk assessment system for the sea ice disasters on fixed jacket platforms. The investigation has great practical relevance and could be potentially used in safety analysis or estimation for structures facing sea ice disasters. The paper is generally well written but several points can be further improved:

- **1) Page 2 line 24: There is an additional "," after 2014.**

Answer:

The additional "." was deleted.

- **2) Figure 1 title: There is an additional "(".**

Answer:

The additional " (" was deleted.

- **3) Figure 2: One should change the order of sub-figure (e) and (d).**

Answer:

the order of sub-figure (e) and (d) was changed.

- **4) Page 5 line 6: The reference "Zhang et al., 2007" should be "Zhang and Li, 2007"?**

Answer:

It was Revised.

- **5) Page 5 line 20: Note the index "H" in "the hazard index (H)" should be italic.**

Answer:

It was Revised.

**6) Page 6: From line 1 to 12, the positions of Equations 1-3 are not in the correct place in the text. One must put the equation directly after the text where the equation is mentioned.**

Answer:

It was reviesed.

- **7) Page 6, line 16-19: "Sea ice disasters have three major risk sources … and four structural risk modes …" The author should give the reference for this.**

Answer:

The reference was given

- **8) In general, the equations are not in the same format. Note the blank spaces in front of the equations are different.**

Answer:

It was reviesed. There is no spaces in front of the equations now.

- **9) In general, many index/sign used in equations are repeated. It is suggested**

**one use different indexes/signs for different parameters or at least give them different subscripts.**

Answer:

The equations were revised. Mainly for the index "H":

(1) hazard index ($H$)

(2) ice thickness ($t$)

(3) height of structure ($H_{str}$)

- **10) Page 7, line 6-7: Note the subscript of "F_H". Should "H" be italic or not?**

Answer:

It should be italic.

- **11) Page 7, line 17: "After floe is applied on a structure and then broken, if broken ice is not removed in time due to structural blockage, ice accumulates". "floe" is a typo? Also one must rephrase the whole sentence.**

Answer:

Floe means big floating ice sheet.

- **12) Table 2: It should be "MPa".**

Answer:

It was reviesed.

- **13) Page 11, line 7: "… are respectively set to be 1 and 2" why does the author use these parameter values? Any reference?**

Answer:

Ice force on single leg platform equals to one time of ice force on one leg, that was "1"; Ice force on four-leg (2 times 2) equals to two times of ice force on one leg when the ice coming direction was the same with the construction of platform (with the lowest total ice action), that was "2".

Reference was added "Liu, Y.: Research on Dynamic Analysis and Structural Lectotype of Ice-resistant Offshore Platforms, Dalian University of Technoloty. Doctoral thesis, 2007."

- **14) Page 11, line 24: "Ka" is not in the right format. Also one must provide reference why 0.5 is selected for main platforms and 1.0 is selected for auxiliary platforms.**

Answer:

The format of "$K_a$" of reviesed.

the structural function also directly affects the risk level. For example, there are many devices on oil production platforms. The design of manned platforms should pay attention to personnel comfort and their risk is relatively high. Unmanned platforms have a low risk. Kb is the structural function coefficient and its values for manned central platforms, unmanned central platforms, and auxiliary function platforms such as the bollard are respectively set to be 1.5, 1.2, and 1.0.

The ice-induced vibration value M3 is expressed as:$M3=V1*V2*V3*V4^{0.5}$.

By compare contribution of four parameters, which are overturning index V1, dynamic indexV2, ice-induced vibration indexV3, and function indexV4= Kb), The value of V4 made smaller contribution to risk results, so the 0.5 exponent of proposed. This conclusion has been validated by experts, and the conclusion was published in postdoctoral research report" Xu, N. Research on Critical Issues of Sea-ice Disaster Risk Assessment and Prevention

Strategy. 2014".

- **15) Page 12, line 4: "V1" correct in format? Line 7 the reference should be "Yue et al., 2007b". Line 8: Acceleration "A" should be italic.**

Answer:

All are revised

- **16) Page 12, line 22: "… such as the bollard are respectively set to be 1.5, 1.2, and 1.0" why? Any reference?**

Answer:

the structural function also directly affects the risk level. For example, there are many devices on oil production platforms. The design of manned platforms should pay attention to personnel comfort and their risk is relatively high. Unmanned platforms have a low risk. Kb is the structural function coefficient and its values for manned central platforms, unmanned central platforms, and auxiliary function platforms such as the bollard are respectively set to be 1.5, 1.2, and 1.0.

This conclusion has been validated by experts, and the conclusion was published in postdoctoral research report" Xu, N. Research on Critical Issues of Sea-ice Disaster Risk Assessment and Prevention Strategy. 2014".

- **17) Page 12, line 25: "M3" is not in the right format.**

Answer:

It was reviese

- **18) Page 19, line 17: what are "level ice" and "rafted ice" and the difference between their parameters? Can one give more detailed explanation?**

Answer:

The following sentence was added in the manuscript "such as the range of ice thickness, the value of ice strength, and the their weight for the risk value under different risk modes."

- **19) Page 20, line 15: The author must keep the reference format consistent. For example, for the page number the sign "1-70" is in different formats.**

Answer:

It was revised.